# The Role of the Flavivirus Replicase in Viral Diversity and Adaptation

**DOI:** 10.3390/v14051076

**Published:** 2022-05-17

**Authors:** Haley S. Caldwell, Janice D. Pata, Alexander T. Ciota

**Affiliations:** 1The Arbovirus Laboratory, Wadsworth Center, New York State Department of Health, Slingerlands, NY 12159, USA; haley.caldwell@health.ny.gov; 2Department of Biomedical Sciences, State University of New York at Albany, School of Public Health, Rensselaer, NY 12144, USA; janice.pata@health.ny.gov; 3Wadsworth Center, New York State Department of Health, Albany, NY 12208, USA

**Keywords:** arboviruses, quasispecies, replication complex, NS3, NS5, RNA dependent RNA polymerase, fidelity

## Abstract

Flaviviruses include several emerging and re-emerging arboviruses which cause millions of infections each year. Although relatively well-studied, much remains unknown regarding the mechanisms and means by which these viruses readily alternate and adapt to different hosts and environments. Here, we review a subset of the different aspects of flaviviral biology which impact host switching and viral fitness. These include the mechanism of replication and structural biology of the NS3 and NS5 proteins, which reproduce the viral genome; rates of mutation resulting from this replication and the role of mutational frequency in viral fitness; and the theory of quasispecies evolution and how it contributes to our understanding of genetic and phenotypic plasticity.

## 1. Introduction

Flaviviruses, which cause severe morbidity and mortality in humans around the world, are single-stranded positive sense RNA viruses typically vectored by arthropods including mosquitoes and ticks. Flaviviruses are found on every major continent except Antarctica, and infection in humans results in a range of clinical outcomes including debilitating neurological or hemorrhagic complications. Many of these pathogens are emerging or re-emerging and have resulted in significant epidemics in recent years. West Nile virus (WNV) emerged in 1999 in the Western Hemisphere, causing an epidemic resulting in massive avian fatalities and encephalitis in humans. Due to large populations of competent *Culex* spp. mosquitoes and passerine birds, West Nile has since become endemic in the USA resulting in an estimated 7 million human infections [1,2,3]. Similarly, Zika virus (ZIKV) has recently emerged, causing epidemics in Micronesia in 2007, French Polynesia in 2013–2014, and ultimately in the Western Hemisphere beginning in 2016. ZIKV caused an estimated 440,000–1,300,000 infections in the Western Hemisphere alone. These recent epidemics of ZIKV demonstrated additional complications not previously associated with infection, including Guillain–Barré syndrome in adults and congenital Zika syndrome in infants [4]. Dengue virus (DENV) is already endemic in areas containing more than a quarter of the world’s population and results in an estimated 400 million infections per year. As with the majority of flaviviruses, most infections (~80%) remain asymptomatic or subclinical with 20% resulting in illness and 5% progressing to severe disease which, for DENV, can include hemorrhaging or shock [5,6]. Yellow fever virus (YFV), although vaccine preventable, is also re-emerging and can cause a hemorrhagic fever which can progress to kidney failure. Prior to 2016 there were relatively fewer cases recorded worldwide, but cases began to spike thereafter at highly irregular intervals particularly in Africa. Although there is a paucity of data on yellow fever cases, currently there is an estimated incidence of 10,350 per 100,000 people [7]. The reemergence of YFV is due to waning immunity, vaccine shortages, invasion of vectors to urbanized areas, and increasing urbanization [8]. ZIKV, DENV, and YFV differ in transmission cycles from WNV, with *Aedes* spp. Serving as the principal vectors and humans capable of acting as amplifying hosts [4,6]. Although these flaviviruses continue to reemerge and spread, there are currently no approved antiviral drug treatments. Additional vaccine preventable flaviviruses, like Japanese encephalitis virus (JEV), continue to cause periodic outbreaks and significant disease burdens [9].

Although the differences between invertebrate and vertebrate infection have yet to be fully elucidated, the generic processes of infection and replication are relatively well characterized for flaviviruses. Entry is facilitated via the envelope protein (E) which exists as a series of dimers together with the viral membrane protein (M) on the surface of the nearly icosahedral virion [10,11]. Within the virion is the viral genome which is complexed with the capsid protein (C). The E protein recognizes one or more cell surface proteins including but not limited to heparin or other glycosaminoglycans as attachment factors and binds to a host receptor which has not yet been defined for all flaviviruses [12]. Other implicated attachment factors include DC-SIGN, L-SIGN, TIM, and TAM, and heat shock proteins [13,14,15,16,17,18]. There is a paucity of data regarding host-specific receptors in both vertebrate and invertebrate hosts and much of the current work has focused on DENV. Entry occurs primarily via clathrin-mediated endocytosis as noted in WNV, DENV, and ZIKV [9,10,11,12,13,14,15,16,17,18,19,20,21]. This method of internalization requires the E protein to undergo significant conformational changes to bring the viral and endosomal membranes in close proximity with membrane fusion occurring in a pH-dependent manner. The linear viral genome is then released from the endosome, where it is directly translated into a polyprotein with a single open reading frame, with its 5′ type 1 cap aiding in ribosomal initiation [22,23]. The polyprotein is processed by both cellular and viral proteases into individual structural and non-structural proteins [24]. These new proteins co-opt the remaining viral genomes and begin asymmetric replication, with the negative-sense RNA genome acting as template for more positive-sense RNA genomes [25]. Resultant genomes are capped. Assembly and replication occur within the cytoplasm, where viral genomes assemble with E, C, and prM proteins and bud into the ER lumen [26,27,28]. The particle is subsequently acidified, allowing furin to cleave the prM protein to prevent viral fusion during egress [29].

The replication complex for flaviviruses has been extensively studied, however far less is understood regarding the replication-independent roles of the nonstructural proteins. The 7 nonstructural (NS) proteins, including NS1, NS2A, NS2B, NS3, NS4A, NS4B, and NS5, form a membrane-bound replication complex in invaginations within the ER (Figure 1). This is believed to protect assembly and replication processes from the host cell. NS1 anchors the complex to the lumen of the ER, while NS2A serves as a scaffolding protein as does NS4A. While the role of NS4B is less clear, it has been found to interact with NS3 and is essential for replication [30,31,32,33,34]. NS2A and NS4A have also been implicated in replication [31,32]. NS3, its cofactor NS2B, and NS5 form the core of the replication complex with NS5 directly replicating viral RNA and capping it while NS3 unwinds dsRNA intermediates. The activities of both of these enzymes are critical in viral replication and their mechanism of action has been well characterized [30,32]. Here, we review what is known about the role of each of these primary replication complex proteins in flavivirus fitness, virulence and transmission, and identify how further insight into the mechanisms of host-specific replication and evolution could inform novel therapeutics and concepts in virus adaptation.

## 2. Structure and Function of NS3 and NS5

The NS5 is the largest flaviviral protein and is a natural fusion of a methyltransferase (MTase) and an RNA-dependent RNA polymerase (RdRP) (Figure 2A). The C-terminal domain includes the RdRP while the N-terminal includes the MTase domain, identified by the AdoMet-dependent MTase motif as well as a guanylyltransferase, indicated by a GTP-binding site and previously characterized capping mechanisms [37,38]. Methyltransferase activity has been confirmed using a recombinant full length NS5 and truncated NS5 containing only the MTase, and via incorporation of radiolabeled methyl groups [39,40,41]. RdRP activity was ascribed to this domain with similar methodology, utilizing a recombinant NS5 fusion protein with glutathione S-transferase with RdRP activity and performing a radiolabel assay to determine incorporation of the label in the newly synthesized RNA [42]. Guanylyltransferase activity via the detection of the GMP-enzyme intermediate when GTP is used as a substrate, the GMP moiety can be transferred to the diphosphate end of the RNA transcript [43]. NS5 is a primer-independent RdRP, meaning it is able to initiate replication de novo. This initiation is carried out using the priming loop, an extension of the thumb domain. The initial step is binding of two nucleotides, including a purine, at the priming site, as well as binding of the template RNA, which is mediated by genome cyclization and occurs due to complementary sequences in the 3′ and 5′ UTRs Figure 2B [44]. The RdRP recognizes a conserved stem loop region within the 5′ UTR which acts as a promoter [45]. The priming loop serves to act as a platform to stabilize these initial steps. Two divalent cations bind to the active site for which binding sites are highly conserved in RdRPs. Catalysis results in the generation of a pyrophosphate, a dinucleotide and translocation. The remaining dinucleotide can be elongated via NTP hydrolysis [46,47].

The NS5 RdRP shares a highly similar structure to other polymerases with the domains resembling a cupped right hand [51,52]. Within the three structural domains: fingers, palm, and thumb, there are seven conserved motifs (A-G) (Figure 2A-C) [37,53,54]. This hand can adopt an open and closed conformation, mediated by movements of motifs F and G [52]. The open conformation exposes the active site during initiation and elongation, while the closed conformation clasps around the incoming nucleotide to facilitate addition [46]. The active site consists of three conserved aspartates located in motifs A and D which are critical for catalysis. Motif B contacts the template and incoming rNTP and has a highly flexible loop which may be involved in translocation [55]. Motif D acts as scaffold for the palm domain and is important during conformation changes. Motif E aids in NTP binding (Figure 2C) [54,56].

The flaviviral NS3 protein contains a C-terminal helicase (NS3h) and an N-terminal serine protease. The helicase belongs to superfamily 2 [57] which is represented by the well-characterized hepatitis C virus (HCV) NS3 [58]. Although not a flavivirus, HCV is a member of the *Flaviviridae* family and the *hepacivirus* genus and shares many similarities with members of the *flavivirus* genus (including WNV, DENV, YFV, ZIKV). The minimal qualification for a superfamily 2 helicase is ATP-dependent directional translocation [57,58]. For flaviviruses, helicases are directional, translocating in the 3′–5′ direction along the tracking RNA during unwinding and can traverse breaks in the phosphodiester backbone [59,60]. NS3h is composed of three subdomains known as 1, 2, and 3 or alternately, N, C, and 3. Subdomains 1 and 2 are conserved across all SF2 helicases where they contain the characteristic Rec-A-like folds of the helicase core and form a deep groove housing the nucleotide binding pocket, in motif 1, and the RNA binding sites [57,58,60]. Motif II binds Mg^2+^ cations required for ATP catalysis [61]. Despite its stated role in unwinding dsRNA during replication, NS3h has been found to translocate and bind to both DNA and RNA, possibly due to the extensive contacts to the phosphodiester backbone of the tracking strand which stabilizes unwinding [62,63,64]. These contacts are not DNA or RNA specific and are mediated instead via water molecules or directly via subdomains 1 and 2 and motifs 1a, TxGx, 4 and 5 [58,59]. However, when NS2B and NS3 are bound together, DNA unwinding is repressed [65]. This difference in selectivity may reflect the different function required in different aspects of the viral life cycle, with NS2B-NS3 being required during polyprotein processing and NS3/NS5 complex being required in ER invaginations to aid in replication [66,67]. Motif VI is involved in altering the conformation of NS3h to closed conformation of NS3 where the substrate is gripped [58,67]. NS3h activity is powered by ATP binding, which is coordinated via motifs I, II. Binding weakens nucleic acid interaction via domains Ia, TxGx, 4, and 5, allowing translocation and unwinding. Hydrolysis of ATP via motifs I, II, and VI triggers rebinding to the nucleic acid in a forward motion while motif III coordinates these processes [57,58] acting as a linker between motifs I and II [68].

In addition to its helicase and NTPase activities, NS3h acts a triphosphatase on positive sense RNA genomes to begin the coordinated process of capping at the 5′ end [69]. The triphosphatase removes a single phosphate group from the backbone of the positive-sense RNA strand, resulting in a diphosphate [39]. At this point, the NS5 methyltransferase (MTase) located at the N terminus of NS5, dephosphorylates GTP into a GMP moiety and transfers it to the 5′ dephosphorylated end of the positive sense RNA to be capped via its guanylyltransferase activity [70]. The core of the MTase consists of beta strands alternating with alpha helices, and contains a SAM binding pocket, RNA binding site, GTP pocket, and K-D-K-E motif at the active site (Figure 2D) [41,71]. The MTase domain also contains two other distinctive enzymatic functions: a guanine-N7-methyltransferase and an RNA 2′O-methyltransferase at the N-terminus [39,41]. Both processes utilize S-adenosyl-l-methionine (AdoMet) as a methyl donor, converting it into S-adenosyl-l-homocysteine. These methyl groups are transferred sequentially onto the N7 atom of the cap guanine and the 2′-hydroxyl group of the ribose moiety of that same RNA nucleotide (Figure 2D) [39,41,67,70]. This results in a type 1 cap structure that is identical to host mRNA, allowing for efficient translation by host machinery and increased stability as well as immune evasion [72,73,74,75]. Mutations that impair MTase activity decrease or abolish replication, indicating the importance of capping and the MTase in viral replication [41].

The role of NS3 in host-specific adaptation has been examined for WNV, with the two most prominent examples being the naturally occurring substitution T249P and the laboratory-derived S411A. T249A is a well-studied mutation that was positively selected when WNV first arrived in North America in 1999 and has been linked to increased virulence in American crows, although this phenotype is not consistent across avian species [76,77]. Mechanistic studies of this mutant NS3 revealed increased ATP activity but similar helicase activity, which failed to explain the altered phenotype of strains with this substitution. Temperature studies indicated increased viral load in avian fever conditions, and further mechanistic studies using altered temperatures may help to elucidate how mechanistic differences reflect phenotypic differences [77]. Experiments with hyperactive Kunjin virus containing the S411A NS3 mutation resulted in decreased viral infection of cells, increased mortality and dissemination in mosquitoes, as well as decreased infection. These results in part can be explained by the hyperactive helicase activity caused by the mutation, which could increase cytopathogenicity and decreased mosquito survival [78].

Although the chemical processes of flaviviral replication and capping are well characterized, the coordination of each of these steps remains elusive. Flavivirus NS proteins clearly interact via the replication complex and the NS3 and NS5 enzymes via direct contact [79], yet the precise role of each domain and protein in replication is unknown. While the MTase has been extensively studied in RNA capping, its role in RNA replication is unclear. However, deletion of the MTase domain significantly decreases viral RNA synthesis, suggesting other roles for the MTase in replication [52,80]. How the replicated RNA is passed between the NS3h, RdRP and MTase for complete genome replication is largely unknown. While crystal structures for full length NS3 and NS5 have been solved for many flaviviruses [35,36,51,52,81], none have been solved for NS3 bound to NS5. The orientation of the MTase relative to RdRP also differs between crystal structures. Several models have been posited for how RNA is replicated and capped. Crystallization of the JEV NS5 revealed a small and flexible linker region, which connects the two domains of NS5, as well as a MTase entry site on the opposite side of the NS5 dsRNA exit path [51]. A highly conserved three amino acid sequence (GTR) across flaviviruses is found adjacent to the variable linker. This sequence could serve as a pivot point, allowing for flexible movement of the MTase to different orientations, including one in which the entry site of the MTase is perfectly aligned with the RdRP exit [51]. Subsequent studies demonstrated the importance of both the GTR sequence and the linker. Mutagenesis of the GTR region in DENV severely repressed viral replication [82]. DENV2 NS5 containing either JEV or ZIKV linkers completely abolished viral replication, potentially due to altered flexibility hindering interactions with either the RdRP or NS3 [83]. Later work also supported the idea of the GTR region acting as a hinge; studies resolved a DENV NS5 structure with an altered MTase orientation. Upon aligning the JEV NS5 (PDB 4K6M) to their own structure (PDB 5CCV), Klema et al. found using DynDom that significant bending of the linker region using GTR as a pivot was required to perfectly align the structures [48]. Interestingly, they also crystallized DENV NS5 in a dimer formation where the methyltransferase entry site and RdRP exit are located on the same side and propose that this may be how flaviviral RNA replication and capping is coordinated [48,70]. Although progress has been made in unraveling the journey of a flavivirus RNA during replication, more studies are required to precisely define how the RNA is processed and how the initial step of capping is performed by NS3.

## 3. Fidelity

RNA viruses, including flaviviruses, are notorious for having highly error-prone genome replication relative to bacteria and eukaryotes whose genomes are replicated by DNA polymerases rather than RNA polymerases. The reason behind this discrepancy stems largely from the lack of proofreading displayed by all RNA viruses, which lack exonucleases able to excise mispaired bases (except for coronaviruses). The result is a mutation rate of approximately 10^−4^ mutations per nucleotide copied, or approximately 1 mutation per genome replicated for a typical 11kb flavivirus [84,85,86]. These high mutation rates generate many diverse viral genomes within the course of a single infection, due to the high rate of viral replication that leads to significant viral loads [84]. This diverse array of genomes is often referred to as quasispecies. The composition and breadth of quasispecies (addressed in detail below) have a significant impact on viral fitness and adaptability when new selective pressures are introduced [87,88,89,90,91,92].

Fidelity refers to the accuracy of replication enzymes in faithfully copying genomes, RNA viruses have relatively low fidelity compared to other organisms. This is both beneficial in terms of the generation of diversity and problematic in that these viruses are theorized to be maintained close to error catastrophe, where the accumulation of deleterious genomes due to high mutation rates leads to extinction [93,94]. This notion has been shown experimentally through the use of mutagens, including ribavirin, favipiravir, and 5′fluorouracil (5-FU) in a variety of RNA viruses including flaviviruses [95,96,97]. Higher concentrations of ribavirin have been shown to result in: (1) Reduced levels of viable virus, (2) reduced specific infectivity, and (3) increased mutation frequency, which correlated with loss of viable virus and lethal mutagenesis. WNV has been driven to lethal mutagenesis by favipiravir when used in vitro, although the precise mechanism of action of favipiravir remains unknown [98]. Thus, error catastrophe can be induced via loss of replication fidelity [97,99].

Perturbation of viral fidelity can thus be a useful tool in studying viral evolution and the importance of viral diversity in fitness and adaptation. Ribavirin has been used extensively to experimentally isolate high-fidelity mutants of viruses including poliovirus, St. Louis encephalitis virus (SLEV), WNV, Coxsackievirus B3, Venezuelan equine encephalitis virus (VEEV), and chikungunya virus (CHIKV) [87,91,100,101,102,103,104]. Interestingly, many of the fidelity variants generated in this manner contained structural changes far from the active site [91,101,104], suggesting that remote sites can greatly impact RdRP fidelity through indirect effects on catalysis. For instance, poliovirus G64S, a high-fidelity substitution isolated through passage in ribavirin, was shown to increase fidelity by stabilizing the triphosphate in a catalytically competent conformation [101,105].

Several fidelity mutants of the picornaviruses have since been identified either by passaging virus in the presence of a mutagen or via structural prediction. While picornaviruses are structurally and biochemically distinct from flaviviruses, these studies provided the basis for our understanding of the phenotypic consequences of altered fidelity for RNA viruses. Foot-and-mouth disease virus fidelity mutants W237F (high fidelity) and W237I (low fidelity) and W237L (low fidelity) demonstrate universally decreased virulence and mortality within mice but not within cell culture, similar to previous studies [106]. This result is likely due to cell culture being poorly representative of host environments [75]. Replication fidelity has been measured by genome sequencing or biochemical assays. Specifically, single nucleotide discrimination of CTP vs 2′dCTP with the high fidelity mutant W237F demonstrated the highest discrimination factor while the low fidelity mutants had the lowest [107]. Low and high-fidelity mutants have also been generated for Coxsackievirus B3, which were characterized using similar assays, as well as single nucleotide incorporation assays which detect correct incorporation via fluorescence. Nine low fidelity mutations were identified which, similar to previous picornavirus fidelity mutants, were attenuated in vivo but not in vitro [102]. Additionally, the lowest fidelity mutants generated were not viable and either did not grow or had new attenuating mutations, reinforcing the notion of error catastrophe. Further studies identified high fidelity mutations including F364Y, which again demonstrated in vivo attenuation [103]. As such, fidelity mutants are largely attenuated in vivo, potentially due to the lack of diversity generated by high fidelity mutants and the proximity to error catastrophe in low fidelity mutants.

The impact of altered fidelity has also been evaluated in arboviruses, including CHIKV, VEEV, SLEV, and WNV, which unlike picornaviruses are multi-host and thus experience unique transmission bottlenecks and additional selective constraints. Initial studies identified fidelity variants in CHIKV following passage in 5-FU [100,108]. The mutation C483Y was shown to be associated with a high-fidelity phenotype determined via sequencing because in vitro biochemical nucleotide incorporation assays do not exist for alphaviruses [100]. Unlike the picornaviral fidelity variants, C483Y, located in the RdRP, demonstrated no attenuation in vitro, although in competition the WT grew significantly better in insect cell lines. This result was reflected in vivo, with significantly lower viral loads in the bodies and legs of infected mosquitoes. Similarly, during infection of a mammalian host, significant attenuation in viremia and tissue tropism of C483Y was identified. Low fidelity mutants were also generated via substitution at the 483 position [108]. Unsurprisingly, these variants produced more defective genomes than WT and were also attenuated in mammalian hosts in each tissue type (muscle, blood, brain, and liver). Unlike the high-fidelity mutant, low fidelity mutants were significantly attenuated in mosquito cell culture when one-step growth kinetics were measured. In mosquito hosts, reversion of the low fidelity mutants occurred, consistent with attenuation driving selection against this phenotype. While increased viral diversity could be preferable within these hosts due to selective pressure from the mosquito RNA interference response (RNAi), which is based on sequence similarity [109], there is clearly a limit to this benefit. Further studies on alphavirus fidelity with VEEV identified a low fidelity variant following passaging in the presence of 5-FU. Unlike previous studies on CHIKV, the low fidelity VEEV variant was more virulent than the WT and showed no attenuation in tissue tropism. As different electroporation pools (used to inject mutant virus constructs into host cells, allowing them to replicate and serve as a stock of the mutant virus in question) generated from the low fidelity mutant were shown to have significantly different impacts on virulence, this study highlighted the importance of the individual composition of quasispecies and the tremendous impact they can exert on viral fitness [110].

The impact of fidelity on flavivirus fitness and host switching has also been examined for WNV and SLEV. WNV fidelity mutants were generated with selection for ribavirin resistance and included a high-fidelity mutant containing V793I and G806R NS5 substitutions located in the thumb subdomain of the RdRP, and a low fidelity mutant, with a T248I substitution in the methyltransferase. The location of these mutations offers unique insight into the role of remote interactions in regulating flavivirus fidelity. The WNV RdRP mutations demonstrate the influence of the priming loop on polymerase fidelity and the low fidelity mutant demonstrated, for the first time, that structural alteration of the flavivirus methyltransferase can have a significant impact on polymerase fidelity. These fidelity phenotypes were confirmed via in vitro primer extension assay, which examines the ability of NS5 to incorporate each of the nucleotides accurately during elongation. These studies confirmed deep-sequencing data demonstrating the fidelity can be mispair specific. The fidelity differences were most pronounced for A:C misincorporation, suggesting specific structural perturbations could be associated with unique mutational biases and evolutionary trajectories. In vitro cell culture studies revealed no attenuation in vertebrate cells, but attenuation in mosquito cells as was seen with CHIKV fidelity mutants. Vector competence of *Culex* spp. mosquitoes for WNV fidelity mutants was shown to be significancy impaired, with highly attenuated infectivity and no evidence of virus transmission [104]. Studies with the SLEV high fidelity mutant, attributed to a E416K RdRP substitution, additionally support the previous results demonstrating attenuation in mosquito but not mammalian cells. Interestingly, this attenuation was limited to mosquito cells with an intact RNAi response, providing further evidence that a limited capacity for RNAi evasion likely contributes to the decreased fitness of high-fidelity variants in mosquitoes [91]. RdRP fidelity also greatly impacts viral evolution and viability during host switching in vitro. During 20 sequential passages alternating from mosquito to avian cells, only the wildtype survived all attempted passages and maintained a higher viral load, demonstrating that wildtype fidelity is fine-tuned for host switching and perturbing fidelity likely decreases the capacity for long-term maintenance with host alteration [92].

Despite the utility of fidelity mutants in probing the relationships between viral diversity and host adaptations, significant issues plague our understanding of these interactions, which stem both from different methodologies used to define fidelity and conflicting results in unique experimental systems and hosts. Early methods for defining fidelity consisted of sequencing small regions of the genome to examine mutation frequency, which fail to examine diversity across the entirety of the genome and is limited by the uncertain influence of evolutionary pressures and process errors of sequencing platforms. Biochemical assays are more reliable in directly measuring activity and misincorporation by the RdRP but are not available for all viral species. Further, how the context of the host cell environment and the fully intact replication complex could influence polymerase fidelity is not well-defined. Additionally, different biochemical methodologies are used to define different viruses with some utilizing nucleotide incorporation assays and others nucleotide discrimination assays, making attributing differences in phenotype to specific structural variations difficult. Indeed, different results in terms of phenotypic attenuation have even been obtained for the same mutant [87,101,110], highlighting the complexity of accurately defining these relationships. For an excellent review on the benefits and challenges with studies of fidelity variants see Kautz and Forrester 2018 [106].

## 4. Quasispecies and Host Adaptation

Underlying these fidelity studies is the question of how viral diversity impacts viral fitness, adaptation, and evolution. Viral infections, particularly by RNA viruses, result in a diverse cloud of genomes generated by polymerase misincorporation during replication [111]. This diverse population is referred to as a quasispecies, a notion first described in the 1970s in reference to early life as evolving from “replicons” [112]. These replicons were diverse and stemmed from a master sequence, the replicon with the highest fitness or replicative capacity, which is bound to its closely related neighbors by the inevitability of mutation. Thus, quasispecies theory posits that individual mutants are not the subject of selection but rather the population as a whole [112,113]. This is in contrast to Darwinian notions of evolution, were selection acts on individuals within a population. It also argues for the notion of “survival of the flattest” whereby viral evolution favors mutating toward areas of sequence space where fitness is relatively flat (not large valleys or peaks) so that the quasispecies swarm retains fitness in the face of mutation, frequent bottlenecks, and fluctuating selective pressures [113].

From this initial framework stems several related theories regarding quasispecies behavior, including Muller’s ratchet, the effect of genetic bottlenecks on fitness, and error catastrophe. Genetic bottlenecks within and between hosts are predicted to decrease the fitness of quasispecies due to the loss of diversity and the likelihood that any individual variant will have deleterious mutations. Serial plaque transfers of individual variants to cells have supported the notion that genetic bottlenecks decrease fitness, which is known in population genetics as Muller’s ratchet [114]. Muller’s ratchet predicts that small population sizes of asexual organisms with a high mutation rate will accumulate deleterious mutations in an irreversible manner in the absence of outside sources of diversity. However, if larger populations of quasispecies infect a new selective environment or host, adaptive selection will occur. These ideas have been supported by a number of in vitro passage studies [115,116,117,118]. Linked to this theory is the idea of error catastrophe, the theoretical threshold of replication fidelity below which viruses cannot produce sufficient viable genomes due to the accumulation of deleterious mutations [93,119,120]. Numerous antiviral strategies have utilized mutagenic compounds as treatments for viruses like hepatitis C virus (HCV) [121], a *Hepacivirus* in the family *Flaviviridae*, and human immunodeficiency virus (HIV) [122]. Lethal mutagenesis has also been shown to be a viable mechanism to decrease flavivirus fitness [98,123], which is further supported by reports of attenuation of low fidelity variants as discussed above.

Quasispecies exist as an individual unit of selection and work cooperatively or competitively to collectively constitute the fitness of a virus. This notion has been shown experimentally, where the fitness of individual genomes within a quasispecies have been found to be distinct from the viral swarm [90] and neuroinvasive pathogenesis has been rescued to wildtype levels by the addition of diverse quasispecies [88]. Maintenance of quasispecies diversity allows for the existence of variants uniquely fit to a variety of environments [117,124].

Quasispecies are host-dependent [115], which makes intuitive sense given the highly unique selective pressures which exists between different hosts. The trade-off hypothesis suggests that alternating replication in two taxonomically divergent hosts (vertebrates and not vertebrates) with distinct selective environments can explain the relatively low rates of evolution of arboviruses [125]. Indeed, many host-specific mutations result in a fitness trade-off in divergent hosts, yet broadly adaptive mutations have also been noted in experimental systems. Adaptation of WNV following sequential passage of transmitted virus via mosquitoes was associated with substitutions in the WNV replicase without decreased viremia in an avian model, yet comparative studies with SLEV demonstrate that the extent of adaptive trade-offs may be species-specific, even among closely related viruses [126,127]. Additional studies of WNV adaptation utilizing serial or alternate passage in vivo in chicks or mosquitoes also demonstrated host-specific fitness, yet mosquito-passaged WNV results have been variable, demonstrating that adaptive tradeoffs are not inevitable [128]. The relatively lower diversity of resulting quasispecies after passaging in chicks compared to mosquitoes suggests stronger purifying selection in avian hosts for WNV and SLEV. This notion has been supported via sequencing of naturally occurring WNV [129,130] and experimentally via infection of susceptible bird populations [131]. Although whole genome sequencing of naturally occurring WNV in avian organs demonstrated higher levels of diversity than naturally infected mosquitoes [132]. The increased diversity in mosquitoes may be attributed to density-dependent selection resulting from RNAi evasion [133,134]. Interestingly, higher levels of intrahost diversity in the vector were not identified in the tick-borne Powassan virus, highlighting how selective pressures on flavivirus quasispecies are both host and virus-specific [135].

The existence of quasispecies does not preclude continued evolution of the master sequence. Natural evolution of WNV from the ancestral NY99 genotype to the WN02 genotype, characterized by a single consensus substitution in the envelope gene, V159A, was shown to be driven by superior fitness in mosquito hosts [136]. WNV continues to experience adaptive evolution, as ancestral WN02 strains have begun to be displaced by newer strains including WNV NY10 [137]. Such evolutionary events are not limited to WNV; ZIKV has recently emerged in part due to mutation A188V in NS1, which increases infectivity in *Aedes aegypti*, and S17N in the prM, which likely contributes to fitness and virulence in neural progenitor cells [138,139].

## 5. Conclusions

Although progress has been made in understanding the role of quasispecies and individual mutations in host adaptation, much remains unclear. The most accurate sequencing technologies utilize read lengths that make minority haplotype reconstruction difficult, which impedes the capacity to study mutations in appropriate genomic contexts. In addition, many studies to date have utilized experimental host systems and experimentally derived mutations to study adaptation, which are unlikely to fully reflect the selective pressure of natural systems. Mechanistic explanations for many of the phenotypes attributed to specific mutations is also largely lacking. The nature of quasispecies makes it difficult to study the influence of individual mutations with variable frequencies. Rather, reverse genetics approaches are typically used to generate a given mutant as the master sequence and adaptation is studied in this context. The fact that arboviruses are almost exclusively error-prone RNA viruses can likely be explained by evolutionary selection and the perpetual diversification of flaviviruses is arguably among the most important attributes driving the global success of these pathogens. Our knowledge of the process of flaviviral replication, the organization and function of the flaviviral replication complex and its role in host-specific adaptation has greatly increased. Future studies should aim to more accurately define the role of individual proteins of the replication complex in polymerase function and fidelity during replication in distinct cell types, utilize improved sequence and single cell technologies to expand our understanding of quasispecies structures, and focus on phenotypic studies in natural hosts.

## Figures and Tables

**Figure 1 viruses-14-01076-f001:**
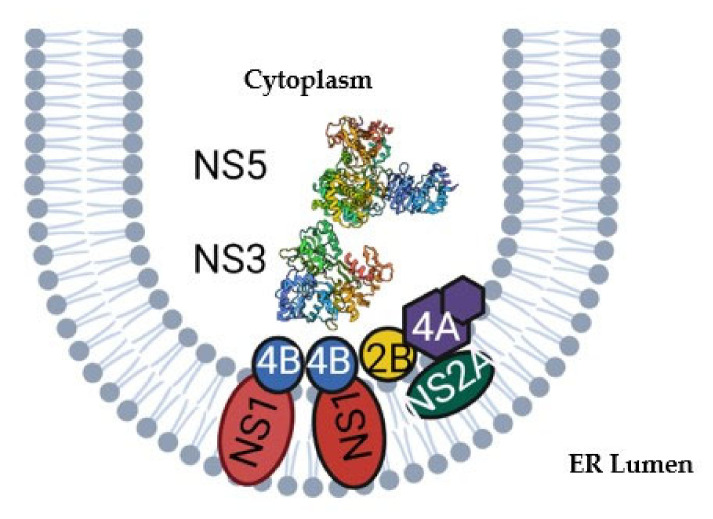
Flaviviral replication complex within an ER invagination. Zika NS5 and NS3 were retrieved from protein data base, codes 5TFR [35] and 5JMT [36], respectively.

**Figure 2 viruses-14-01076-f002:**
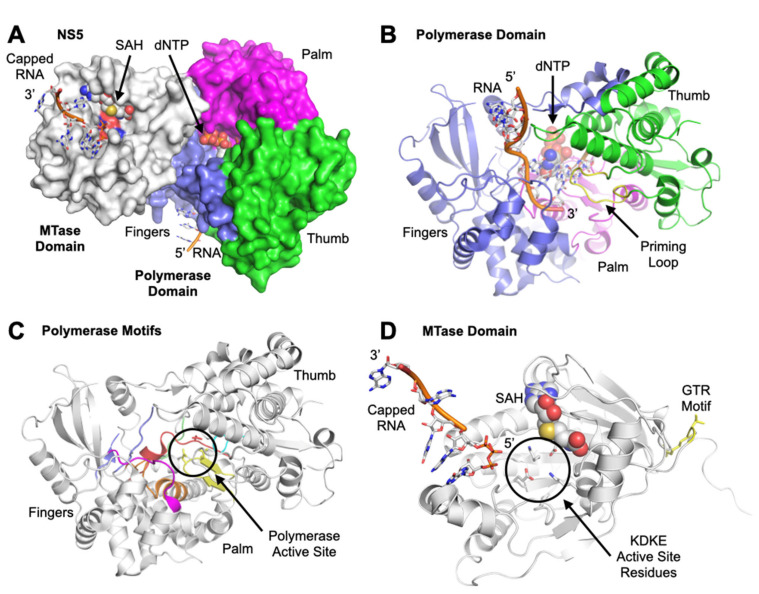
Flavivirus NS5. Dengue structure 5CCV [48] was used to model NS5. (**A**) View of entire NS5 including MTase (white) and RdRP, which is colored by domain. The thumb is green, priming loop yellow, palm pink, and fingers blue. The RNA modeled in the RdRP was manually superimposed using polio RdRP structure 3OLB [49]; the nucleotide is shown in sphere representation. The active site residues exposed on the surface of the MTase are colored by atom type (oxygen, blue; nitrogen, red) while the RdRP active site is not visible in this view. The SAH bound in the SAM binding domain of MTase are shown as spheres. The capped RNA was modeled from Dengue MTase structure 2XBM [50] by aligning the MTase domain to that of the 5CCV structure. (**B**) open view of the RdRP active site containing a newly added CTP (shown as spheres). The RNA backbone is colored orange and the bases are shown as sticks, with carbon atoms colored white, oxygen red, and nitrogen blue. (**C**) RdRP shown in the same orientation as in B, but colored by sequence motif: Motif A is red, B orange, C yellow, D lime, E cyan, F slate blue, and G magenta. The active site aspartates are shown as yellow and red sticks. (**D**) View of the MTase containing the active site, bound SAM shown as spheres and capped RNA shown as in the RdRP. Highly conserved GTR residues are shown in yellow sticks, and active site KDKE residues are shown as sticks colored by atom type as in B.

## Data Availability

Not applicable.

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
