# Peer review of "The Role of the Flavivirus Replicase in Viral Diversity and Adaptation"

_viruses, 2022, doi:10.3390/v14051076_

Round 1

Reviewer 1 Report

This paper by Caldwell et al offers a comprehensive review on the role of flavivirus polymerases as a source of diversity and virus adaptation. The manuscript may be of major interest to those working in the fields of flavivirus research, virus polymerase structure and fidelity, viral quasispecies or antiviral approaches based on lethal mutagenesis. I support without reservation its publication in Viruses once the authors have addressed some minor suggestions listed below:

  1. The manuscript includes only one figure. The paper could greatly improve if a couple of additional figures are added. I’d suggest an illustration of a flaviviral polymerase showing the 7 conserved motifs and highlighting the position of the catalytic site as this is described within the article. The structure of NS3 could also be shown if the authors deem it of interest to the readers. Some other figures depicting other topics mentioned in the text could also be included.
  2. Lines 244-246. I’d include here a sentence mentioning some of the recent studies with favipiravir (T-705, avigan, etc). Several flaviviruses, including West Nile, Zika, Usutu, etc are very sensitive to increases in the mutation frequency elicited by favipiravir (lethal mutagenesis).
  3. Please cite some of the studies published by Cristina Escarmis et al on plaque to plaque transfers and the Muller’s ratchet

Typos:

  • line 90. ….are critical
  • lines 126 and 128, conformation instead of confirmation
  • line 336, a higher viral load / higher viral loads

Author Response

Thank you for spending time reviewing:  "The role of the flavivirus replicase in viral diversity and adaptation". All minor grammatical changes have been made. To address the first major point, another figure has been added which shows all of the domains, motifs, active sites and more of the flavivirus NS5 (both MTase and RdRP).

Regarding the second major point a sentence and references have bee inserted regarding favipiravir (lines 264-266)

Thank you for pointing out more relevant literature the authors initially missed! References to escarmis have been included on lines 399 and 402 of the manuscript.

All major changes are highlighted.

Thank you.

The authors

Reviewer 2 Report

   This is a straightforward and well-organized review on flavivirus replicase enzymes/factors and viral evolution/adaptation.  It serves as an excellent primer for this area of the field and I believe that it will be significant value to researchers.  I do have a few suggestions to polish the manuscript:

Major Point:

  1. The structure/function of NS3 and NS5 section would be much improved by the inclusion of figures to make the details that are presented in the text more approachable and digestible to the general reader.

Minor Points:

  1. Line 34 Intro:  change ‘Zika caused…’ to ZIKV caused….’
  2. Line 75-76: The sentence: 'Resultant genomes are capped while other genomes serve to create more protein components.’ is unclear as it implies that uncapped genomic RNAs are translated.
  3. Line 118: change ‘act as platform’ to ‘act as a platform’

Author Response

Thank you for spending time reviewing:  "The role of the flavivirus replicase in viral diversity and adaptation". All minor grammatical changes have been made, and the sentence regarding capping and genomes has been clarified. To address the major point, another figure has been added which shows all of the domains, motifs, active sites and more of the flavivirus NS5 (both MTase and RdRP).